# Modulating Cognition-Linked Histone Acetyltransferases (HATs) as a Therapeutic Strategy for Neurodegenerative Diseases: Recent Advances and Future Trends

**DOI:** 10.3390/cells14120873

**Published:** 2025-06-10

**Authors:** Huong Anh Mai, Christina M. Thomas, Gu Gu Nge, Felice Elefant

**Affiliations:** Department of Biology, Drexel University, Philadelphia, PA 19104, USA; am4829@drexel.edu (H.A.M.); cmt397@drexel.edu (C.M.T.); gn345@drexel.edu (G.G.N.)

**Keywords:** neuroepigenetics, histone acetyltransferases (HATs), histone deacetylases (HDACs), neurodegenerative diseases (NDs), histone code, learning and memory, cell death, synaptic plasticity

## Abstract

Recent investigations into the neuroepigenome of the brain are providing unparalleled understanding into the impact of post-translational modifications (PTMs) of histones in regulating dynamic gene expression patterns required for adult brain cognitive function and plasticity. Histone acetylation is one of the most well-characterized PTMs shown to be required for neuronal function and cognition. Histone acetylation initiates neural circuitry plasticity via chromatin control, enabling neurons to respond to external environmental stimuli and adapt their transcriptional responses accordingly. While interplay between histone acetylation and deacetylation is critical for these functions, dysregulation during the aging process can lead to significant alterations in the neuroepigenetic landscape. These alterations contribute to impaired cognitive functions, neuronal cell death, and brain atrophy, all hallmarks of age-related neurodegenerative disease. Significantly, while age-related generation of DNA mutations remains irreversible, most neuroepigenetic PTMs are reversible. Thus, manipulation of the neural epigenome is proving to be an effective therapeutic strategy for neuroprotection in multiple types of age-related neurodegenerative disorders (NDs) that include Alzheimer’s disease (AD), Parkinson’s disease (PD), Amyotrophic lateral sclerosis (ALS) and Huntington’s disease (HD). Here, we highlight recent progress in research focusing on specific HAT-based neuroepigenetic mechanisms that underlie cognition and pathogenesis that is hallmarked in age-related NDs. We further discuss how these findings have potential to be translated into HAT-mediated cognitive-enhancing therapeutics to treat these debilitating disorders.

## 1. Introduction

Within the nucleus of eukaryotic cells, the chromatin environment is organized into nucleosomes, which comprise DNA wrapped around a histone octamer core consisting of two copies each of histone proteins H2A, H2B, H3, H4 [1,2]. Such compact chromatin organizations present a major physical hurdle that transcriptional machinery must overcome to initiate gene transcription in vivo. Chromatin packaging is tightly controlled by tissue and site-specific covalent epigenetic post-translational modifications (PTMs) of the core histone proteins that can trigger chromatin packaging changes to activate or repress genes. Thus, epigenetic PTMs elicit heritable phenotypic variation without altering the nucleotide sequence that comprise genes [3]. Epigenetic PTMs include acetylation, methylation, phosphorylation, ubiquitination, sumoylation, and ADP-ribosylation [4]. Histones are often concurrently modified on several residues that allow for an active interplay between histione PTMs and DNA methylation, thus creating diverse combinatorial possibilities for transcriptional control [5]. These modifications not only can modulate chromatin structure, but can also serve as distinct docking sites for other protein complexes that activate or repress transcription [6,7]. Regulation of these PTMs is governed by enzymes that function as “writers” to add specific PTMs to histones, recognized by “readers” that comprise regulatory proteins that control chromatin packaging, and removed by enzymes that serve as “erasers”. Importantly, the recruitment of chromatin-interacting proteins often depends not on a single PTM, but rather on a combination of local marks [7]. These chromatin regulatory proteins work in concert to determine the transcriptional programs employed during the cell differentiation process, timing-specific developmental stages, or stably maintained gene control across generations [8].

Recent explorations into the brain neuroepigenome are providing unprecedented insights into the impact of PTMs of histones in controlling gene expression, not only in early brain development but also in adult brain cognitive function [9]. Together, with studies implicating altered chromatin structure and histone PTM patterns in multiple age-related neurodegenerative diseases, these findings have generated intense interest into investigating how the aging process can detrimentally affect the brain neuroepigenome. While age-related generation of DNA mutations is irreversible, significantly, the majority of neuroepigenetic modification marks are reversible. One of the most extensively studied reversible PTM critical for mediating cognition is histone acetylation, which serves to initiate a neural circuitry plasticity process via chromatin control, enabling neurons to adapt their transcriptional responses to external environmental cues [10]. Importantly, many compelling studies support the concept that cognitive ability in the ND brain is not entirely lost and can be recovered by therapeutic reversal of histone acetylation-mediated gene transcription. In one such groundbreaking study, researchers demonstrated that cognitive capacities in the neurodegenerating brain in an AD-associated mouse model are constrained by an epigenetic blockade of gene transcription. Such transcriptional repression is caused by an increase in the histone deacetylation enzyme HDAC2 that triggers inappropriate hypoacetylation of chromatin at synaptic plasticity gene loci, thereby reducing expression [11]. Importantly, reducing HDAC2 buildup reinstates histone acetylation to restore plasticity gene expression, promoting renewal of structural plasticity in the brain and memory restoration. Similarly, medicinal activation of HAT p300/CBP to increase histone acetylation in the ND brain induces neuronal survival and stimulation of neurite growth by upregulation of neuroprotective gene transcription [12,13]. Together, these findings highlight the reversible nature of histone acetylation-mediated neuroepigenetic modification marks and their role in modulating memory-related gene expression. Thus, manipulating the neuroepigenome is proving to be a promising strategy for neuroprotection of multiple types of age-related neurodegenerative disorders (NDs) that include Alzheimer’s disease (AD), Parkinson’s disease (PD), Amyotrophic lateral sclerosis (ALS), and Huntington’s disease (HD) [14,15,16,17]. Here, we will discuss recent research findings that link neuroepigenetic histone acetylation to cognition and pathogenesis hallmarked in age-associated NDs. We then provide insights into how these findings have potential to be translated into effective cognitive-enhancing therapeutic strategies to treat these debilitating neurodegenerative diseases. This review synthesizes peer-reviewed studies from 2000–2025.

## 2. HAT Families in Brain Function and Cognition

One of the most intensely researched PTMs to date that is critical for brain function and cognition is histone acetylation, which regulates neural gene expression by controlling chromatin packaging in neurons [10,11,17,18,19,20,21,22]. This epigenetic modification is governed by family members of histone acetyltransferase (HAT) enzymes that catalyze the transfer of an acyl moiety from acyl coenzyme A to lysine residues on the n-terminal tails of the histones [23]. Due to the rapid and reversible nature of histone acetylation, this modification serves as a highly dynamic epigenetic mechanism that links environmental stimuli to the molecular adaptations underlying experience-dependent behavioral changes. Thus, cognition relies on coordinated neural gene expression programs to achieve synthesis of new synaptic proteins, which is dependent on histone acetylation control of chromatin structure in neurons [24]. Acetylation alters the chromatin state in favor of a more relaxed, open euchromatin conformation that is conducive to transcriptional activation [8,25]. To achieve this, acetylation is believed to neutralize the positive charge on lysine residues, thus reducing the interaction affinity between histones and DNA, allowing for a more relaxed chromatin state [1,26]. Recent evidence points to acetylated lysine residues serving as molecular docking sites for proteins, including bromodomain-containing ATP-dependent chromatin remodeling complexes and components of the transcriptional machinery, that further facilitate relaxation of chromatin structure [25,27]. The relaxed hyperacetylated chromatin conformation is characterized by increased sensitivity to DNase digestion and enhanced flexibility of the nucleosomal DNA, due to reduced conformational hindrance from nucleosomal compaction [28]. This chromatin unwinding allows access of the transcriptional machinery to the regulatory elements of the genes, facilitating their expression. Further, the TBP-associated factor (TAF_II_) also possesses intrinsic HAT activity, enabling the unwinding of regions around specific genes to allow TFIID access to previously repressed genomic regions [29]. Other groups have demonstrated increased access to gene promoter regions via histone acetylation, which facilitates the ability of RNA PolII to initiate transcription at specific genetic loci [29]. Based on sequence similarity, substrate preferences, and functions, HATs are divided into distinct families. The three most notable HAT families that are required for brain function and cognition are GCN5/PCAF, MYST, and CBP/p300 [30]. Their functional roles in the learning and memory process are discussed in the following sections (Figure 1).

**CBP/p300 Family.** CBP (also known as CREB binding protein or KAT3A) and p300 (also known as EP300 or KAT3b) are members of the CBP/p300 (also known as KAT3) HAT superfamily [31]. These HATs share highly similar sequence and structural organizations with overlapping yet distinct roles in brain function. They contain protein interaction domains, including the nuclear receptor interaction domain (RID), the CREB and MYB interaction KIX domain, the cysteine/histidine regions (CH1 and CH3), and the interferon response-binding domain (IBiD) [4]. These HATs also contain a histone acetyltransferase domain, a bromodomain that interacts with acetylated lysine residues, and a plant homeodomain (PHD)-type zinc finger motif that is a part of the integral enzymatic core of the acetyltransferase domain. [32]. CBP and p300 were each initially identified in protein interaction screens for CREB (cAMP response element) and E1A, respectively. Several key studies revealed that both CBP and p300 are expressed ubiquitously in the cortex and hippocampus [12]. Although both CBP and p300 have overlapping expression patterns, their distinct roles in memory formation arise from the different protein partners they interact with and the preferential histone lysine residues they target. The first evidence that CBP played a role in cognition was revealed by the discovery that Rubinstein–Taybi Syndrome (RTS), a severe neurodevelopmental disorder accompanied by mental retardation, was caused by mutations in the gene encoding CBP [33]. Subsequent studies revealed that mice with a haploid knockout for CBP (cbp+/−) showed reduced histone acetylation and impairment of hippocampus long-term potentiation (L-LTP) and long-term memory (LTM) [34]. Notably, CBP’s HAT function was necessary for these functions, solidifying a critical role for CBP HAT function in cognition [35]. Additionally, HATs such as the E1A-binding protein p300 (p300) are shown to be critical for memory processes [36,37]. To clarify specific roles of CBP and p300 in brain function, studies have utilized p300 and CBP KIX mutant mice and demonstrated that p300 and CBP play different roles in motor-skill learning [38]. In contrast, studies using HAT mutant p300 and CBP mutant mice revealed that both HATs are specifically necessary for long-term recognition and contextual fear memory [37]. More recently, the CBP KIX domain has also been shown to regulate long-term memory and circadian activity in mouse models [39]. Thus, CBP and p300 appear to play functionally overlapping roles in some but not all types of memory processes.

**MYST Family.** The MYST family compromises five HATs: Tip60 (KAT5), MOZ (KAT6A, MORF (KAT6B), HBO1 (KAT7), and MOF (KAT8) [40]. MYST family members each contain a MYST region of about 240 amino acids, a canonical acetyl-CoA binding site, and a C2HC-type zinc finger motif [41]. They also possess a chromodomain involved in protein–protein interactions and, more recently discovered, in neural RNA interactions [40]. Tip60/KAT5 (Tat-interactive protein-60 kDa) is one of the best characterized MYST family HATs concerning its role in cognition, providing substantial insights into histone acetylation-mediated gene control required for neural function in learning and memory [22]. Tip60 is a member of the Tip60 protein complex shown to possess not only HAT activity, but also ATPase, DNA helicase, and DNA binding activities [27]. In an expression analysis of the 18 mammalian HATs in the learning and memory hippocampal CA1 region in adult mice, Tip60 was identified as the second highest expressed HAT [42]. Similarly, Tip60 is robustly expressed in the *Drosophila* mushroom body, a brain region functionally analogous to the mammalian hippocampus and essential for learning and memory, supporting species conservation of Tip60 brain localization [22]. A plethora of research using the *Drosophila* model system has defined a critical role for Tip60 in multiple neuronal brain functions. For example, reducing Tip60 HAT levels within the fly mushroom body causes significantly shorter mushroom body lobes in the adult fly brain, indicative of defects in axonal outgrowth [22]. Further, loss of Tip60 HAT activity in motor neurons causes axonal transport stalling. Additionally, reduction of Tip60 in the small lateral neuron ventral (sLNv) that regulate sleep/wake cycles in the fly causes repressed axonal outgrowth [43,44]. Functional consequences of these Tip60-mediated defects include impairment in short-term memory, sleep/wake cycles, and locomotor activity [16,22,43]. Additionally, Tip60 HAT activity has been shown to regulate proper synaptic bouton expansion at the *Drosophila* neuromuscular junction through modulation of the synaptic microtubule cytoskeleton [45]. Although the identity of the genes responsible for the morphological and molecular changes underlying learning and memory remain incompletely characterized, advances in high-throughput sequencing technology have begun to uncover epigenetically regulated genes involved in synaptic plasticity and neuronal development. For instance, microarray analysis of Tip60 HAT-mutant flies revealed that Tip60 regulates the expression of genes crucial for neuronal development and synaptic function [46]. Further studies have since revealed that several synaptic plasticity genes are bona fide direct targets of Tip60, and the loss of Tip60 activity in mutant flies results in reduced Tip60 chromatin enrichment and synaptic gene repression [47]. Moreover, chromatin immunoprecipitation sequencing (ChIP-Seq) analysis in the *Drosophila* brain reveals that Tip60 target genes are enriched for functions in cognitive processes and, accordingly, key genes representing these pathways are dysregulated in the Tip60 HAT mutant fly brain [22]. Similarly, Tip60 (KAT5) reduction in the forebrain of adult mice causes substantial transcriptional impairments with concomitant neuronal cell loss in the hippocampal CA1 region that is accompanied by mild memory impairment [48]. Additional studies revealed a role for Tip60 within the hippocampus during strong memory training [49]. More recently, studies characterizing dysregulated gene sets (DEGs) in cKO Tip60 mice identified significant overlap with DEGs in AD patients, highlighting the importance of these genes in neuronal function and AD [50]. Together, these studies support a critical role for Tip60 in regulating transcriptional networks required for neuronal function and cognition in both *Drosophila* and mammalian systems.

While fewer studies have investigated potential roles for MOZ(KAT8), MOF(KAT6B), and HBO1(KAT7) in brain function, each of these HATs are expressed in adult mouse hippocampus, and their knockdown causes defects in neuronal progenitor cell function and repression of neuronal-patterning gene expression [15,51]. MORF (KAT6B) is also highly expressed in both the embryonic and adult brain and thought to function in adult neurogenesis [51]. More recently, sociability, learning, and memory were shown to be impaired in MORF (KAT6B) haploinsufficiency mice, supporting a role for this HAT in cognitive function as well [52]. Additionally, gene expression analysis in MORF (KAT6B) haploinsufficiency mice revealed reduced expression of genes required for brain development, and neuron differentiation and expansion [53]. Additional studies investigating MOF (KAT8) identified embryonic lethality in mice with a conditional MOF deletion that was caused by neural inflammation with metabolic impairment of neuronal cells [54]. Lastly, research focused on HBO1 (KAT7) identified a novel cellular pathway where KO of neurite outgrowth inhibitor-B receptor (NgBR) inhibits KAT7, causing inhibition of downstream transcription and regulatory factors that trigger neuronal damage [55].

**GNAT Family.** The Gcn5-related N-acetyltransferases (GNAT) HAT family contains two primary representative members, GCN5/KAT2A and PCAF/KAT2B, which each play critical roles in the developing brain [56]. The GNAT family HATs contain a C-terminal bromodomain that can bind specifically to acetylated lysine residues and have up to four conserved motifs (A-D) that are within the catalytic HAT domain [57]. Gcn5 has been shown to be the most highly expressed of the 18 HATs present in the learning and memory hippocampal CA1 region in adult mice, and was identified as one of the HAT genes that is upregulated in response to fear-conditional training, implicating a critical role for Gcn5 in cognition [42,58]. Indeed, conditional deletion of Gcn5 causes substantial loss of brain mass, resulting in dysregulation of neural progenitor differentiation [58]. Furthermore, elegant studies demonstrated that knockout of *Gcn5/Kat2a* in mice show impaired hippocampal synaptic plasticity and long-term memory consolidation [42]. The study demonstrated that Gcn5/Kat2a controls a hippocampal gene expression network that is highly interconnected, supporting Gcn5/Kat2a as a critical HAT in the control of memory formation [42].

While there are limited studies regarding PCAF function in the brain, research shows that PCAF-dependent epigenetic changes promote axonal regeneration in the central nervous system. The research showed that extracellular signal-regulated kinase (ERK) axonal retrograde signaling triggers PCAF-dependent acetylation at promoters of axonal regeneration genes, and that overexpression of PCAF itself is sufficient to promote axonal regeneration [59]. Moreover, mice carrying a homozygous knockout of the PCAF gene are viable and show short-term memory impairment at 2 months of age, which is the adolescent stage for mice [36]. Interestingly, memory deficits exhibited in mice carrying the PCAF gene KO become exacerbated during aging with mice exhibiting long-term contextual impairments at 6 and 12 months [36]. Together, these studies support the concept that Gcn5 and PCAF each play distinct roles in particular memory formation processes. 

## 3. HAT: HDAC Interplay in Activity-Dependent Cognition-Linked Gene Control

The generation of chromatin histone acetylation by HATs is reversible and is conversely controlled by the antagonistic activity of histone deacetylase (HDAC) enzymes. HDAC enzymes function to catalyze removal of acetyl groups from specific and conserved histone lysine residues, causing a compacted and transcriptionally repressive chromatin environment [60]. Under healthy conditions, maintaining histone acetylation homeostasis within the central nervous system via the interplay between HAT and HDAC action is required for neuroepigenetic gene control involved in neuronal survival and function. This concept is supported by numerous studies showing that altering HAT/HDAC homeostasis in neurons causes detrimental effects. For example, treatment of dopaminergic neural cell culture lines using rat, mouse, and human cell lines treated with the HDAC inhibitor trichostatin A (TSA) triggers apoptosis of neuronal cells [61]. Furthermore, increasing either HDAC6 or HDAC2 levels in neuronal nuclei results in deleterious consequences for transcriptional regulation required for synaptic function. Similarly, increasing CBP HAT levels in primary cerebellar granule neurons (CGN) causes chromatin condensation and cell death [62,63]. Additionally, appropriate levels of Tip60 HAT activity in the *Drosophila* learning and memory mushroom body center in the brain were shown to be crucial for immediate recall memory [44]. Conversely, targeted overexpression of the HDAC2 (Rpd3) and HDAC4 to reduce acetylation levels within the adult *Drosophila* mushroom body resulted in severely impaired long-term courtship memory [64,65]. These findings highlight the importance of appropriate HAT/HDAC levels and balance in promoting neural health and function.

The human brain contains a multitude of neuronal networks that form synaptic connections that change in response to experiences; however, many of these changes are not long-lasting. The basis for long-term memory formation is dependent upon activity-dependent gene control that is mediated by neuroepigenetic histone acetylation-mediated mechanisms. Activity-dependent gene regulation is a fundamental mechanism by which neurons adapt and fine-tune their transcriptional responses to behavioral, environmental, and neural cues to promote sustained neural plasticity and higher-order brain function [66]. Support for this concept was first demonstrated by compelling studies showing that mice subjected to contextual fear conditioning exhibited significantly enriched histone acetylation marks at distinct sites on H3 and H4 with a concomitant increase in multiple learning-associated genes [58]. Indeed, the specific HATs and HDACs that generate distinct chromatin histone acetylation patterns and the mechanisms by which they respond to external stimuli still remains to be determined. The first HAT identified to be critical in activity-dependent gene control is CBP, which was shown to use Ca2+-mediated signaling pathways in response to external neuronal stimuli. Research using primary mouse neuronal cells in culture dosed with potassium chloride (KCl) [67] showed that Ca^2+^ influx promotes transcription at activity-dependent neuronal enhancers via CBP recruitment to these gene loci. Neuronal stimuli also control chromatin histone acetylation by regulating nucleocytoplasmic transport of specific class II HDACs and Tip60 HAT into and out of the nucleus. In DIV rat hippocampal neurons, stimulation with KCl caused the translocation of Tip60 into the nucleus and transcriptional activation of bona fide Tip60 target genes [68]. Additional studies showed that class II HDACs accumulated in the nucleus when neuronal cell cultures were treated with a nuclear calcium-signaling inhibitor, hindering neuronal activation [69]. This regulation of HDAC intracellular localization is critical in mediating histone acetylation and transcriptional activity of neuronal genes that sense and respond to environmental stimuli [20].

Neuronal activity-mediated sensory experiences also have differential effects on synaptic plasticity at excitatory or inhibitory synapses, triggering long-term potentiation (LTP) or long-term depression (LTD), whereby synaptic transmission is up- or downregulated, respectively [10]. Particular forms of LTP and LTD require long-lasting changes in gene expression that is mediated by histone acetylation. In support of this concept, an elegant study using sensory motor neurons of *Aplysia* showed that the coordinated interplay between CBP HAT and HDAC5 regulates histone acetylation levels to promote the switch between LTP and LTD at the same synapses [70]. Intriguingly, the activity-dependent remodeling events that shape the formation of neural circuits during development mirror those underlying changes in synaptic connectivity necessary for learning and memory formation [71]. Experience-dependent neuroadaptation requires an increase in synaptic bouton densities and dendrite extensions, similar to those essential during development [72]. Therefore, it has been hypothesized that the signaling cascades activated by experience-driven changes in cognitive function modulate epigenetically mediated transcriptional control of genes that promote synaptic growth and neuronal homeostasis.

Post-mitotic neurons within the brain have the remarkable ability to shape and refine their synaptic connections in response to ever-changing environmental external stimuli. Multiple studies have shown that environmental enrichment (EE) that includes diversity in housing, play toys, and exercise apparati significantly enhances memory in mice. Indeed, mice raised under EE conditions show changes in the expression of genes in the brain involved in formation of new synapses, strengthening of existing synapses, neurotransmission, and cytoskeletal changes involved in promoting neurogenesis [73]. Elegant work demonstrated the epigenetic mechanisms underlying these beneficial morphological changes by showing that EE conditions trigger hippocampal induction of histone acetylation specifically at marks H3 (K9, K14) and H4 (K5, K8, K12) [74]. Moreover, work with *Drosophila* demonstrated that EE also enhances histone acetylation marks in the fly mushroom body EE-exposed *Drosophila* brains show significant increase in H3K14ac, H4K5ac, and H4K12ac under the EE condition that is mediated by Tip60 HAT, suggesting that the EE-induced histone acetylation induction response is tightly conserved in both flies and mice [75]. Further, work by [76] demonstrated that rats that were housed in new open-field environments that promoted their exploratory nature caused PTM-associated chromatin remodeling mediated by histone H3 Ser10 phosphorylation and H3 Lys14 acetylation with concomitant activation of the intermediate early genes (IEG) c-Fos and zif/268 protein expression within the hippocampal CA1 location in the brain required for learning and memory. Together, these studies support that HAT/HDAC homeostatic balance plays a critical role in mediating EE transcriptional control that contributes to cognition.

## 4. Neuroepigenetic Alterations Drive Multiple Age-Related Neurodegenerative Diseases

Appropriate histone acetylation homeostasis is maintained by the antagonistic activity of HAT and HDAC [17,77]. Disruption of this finely tuned epigenetic balance in the aged brain is characterized by reduced histone acetylation and triggers an epigenetic blockade of transcription. This impairment is associated with detrimental consequences on neural function and has been proposed to be a key early step in age-related cognitive decline and multiple age-associated neurodegenerative diseases. Accordingly, cases of reduced histone acetylation levels are found in the brains of animal models for multiple types of neurodegenerative diseases [17,74,78,79,80,81]. Pharmacological therapies focused on elevating histone acetylation by HDAC inhibition in ND animal models have shown alleviation of cognitive impairments and are thus a topic of intense research [77,79,82]. This section reviews recent findings that reveal an association between impairment of HAT and HDAC balance and function to alterations in the neuroepigenome that contribute to multiple neurodegenerative diseases (Figure 2).

### 4.1. Huntington’s Disease

Huntington’s disease (HD) is a fatal neurodegenerative disorder characterized by motor and cognitive symptoms caused by significantly extended and unstable CAG repeats (polyglutamine or polyQ expansion) that occur within the coding region of the huntingtin (Htt) gene [83]. This expansion results in production of a mutant HTT protein (mHtt) that contains a toxic polyglutamine (polyQ) tract that causes degeneration of the caudate nucleus and the putamen within the basal ganglia of the brain [83]. Such deficits typically appear in adulthood with severity dependent on the length of the polyQ tract. Transcriptional dysregulation plays a central role in HD pathogenesis with alterations being reported in the brains of both HD patients and multiple HD mouse models [84]. Although the precise mechanism underlying HD-related neurotoxicity remains poorly understood, increasing evidence suggests that the expanded glutamine tract within mHtt confers it with new toxic properties that include aberrant interactions with proteins that include HATs required for chromatin acetylation and gene expression [85]. For example, increased HD with expanded CAG repeats was shown to induce redistribution of CBP into inclusions within the nucleus or cytoplasm in multiple cellular models, thereby reducing HAT activity of CBP, causing global deacetylation and cellular death [86]. Additional studies have shown that a polyglutamine peptide that corresponds to exon 1 of Htt (Httex1p) inappropriately interacts with CBP and PCAF HAT domains in vitro, thereby titrating out HAT activity [87,88]. Such reduced histone acetylation in HD patients, mouse, and *Drosophila* HD models causes chromatin hypoacetylation with concomitant transcriptional dysregulation of multiple genes, including those in involved in long-term memory [89,90].

Studies have shown that polyQ-expanded Htt induces ubiquitylation of CBP and its degradation [91,92]. As such, HDAC inhibitor (HDACi) treatments show restoration of decreased histone acetylation levels and alleviate disease pathology in mouse and *Drosophila* HD models. In support of this concept, induction of HTTex1P in cells induces reduction in H3 and H4 acetylation that is reversed by treating cells with pan-HDAC inhibitors that include trichostatin (TSA) and suberoylanilide hydroxamic acid (SAHA) [90]. Further, neurodegeneration of photoreceptor neurons in *Drosophila* induced by Httex1p can be mitigated through treatment with HDAC inhibitors such as sodium butyrate (NaB) and SAHA. Additionally, suppression of class I HDAC Rpd3 and class III HDAC sirtuin 2 (SIRT2), either alone or in combination by genetic and/or pharmacological manipulation, confers neuroprotective effects [93]. Similarly, treatments of SAHA reduce HDAC4 in the cortex and brain stem in an HD mouse model [94]. In a study exploring the effects of HDAC1 inhibitor 4b, researchers found that within fly, cell, and mouse models of HD, HDAC3 was selectively targeted, and the inhibitor was able to suppress HD-related symptoms [95]. More recent studies demonstrate that a selective inhibitor of HDAC3 (selective inhibitor RGFP966) suppressed striatal CAG repeat expansions and prevented cognitive defects in Huntington’s disease mice [96]. Taken together, these research findings highlight the concept that HD is at least in part caused by reduced histone acetylation levels.

### 4.2. Parkinson’s Disease 

Parkinson’s disease (PD) is a progressive neurodegenerative disorder, characterized by degeneration and death of dopaminergic (DA) neurons in the substantia nigra pars compacta (SNc) within the ventral midbrain [97]. Molecular studies have characterized that the pathological hallmark of PD is by misfolded protein aggregation (Lewy body), which is mainly composed of alpha-synuclein (a-SYN) in PD patients [98]. PD progresses through multiple interconnected mechanisms that include mitochondrial dysfunction, energy depletion, oxidative stress, excitotoxicity, protein misfolding and aggregation, disruption of protein clearance mechanisms, cell-intrinsic disturbances, and the prion-like spread of misfolded proteins [99]. Approximately 10–15% of PD cases are familial and known to be caused by genetically inherited mutations in PD-associated genes such as SNCA, Parkin, and LRRK2, and likely genes that have yet to be identified. Unfortunately, most PD cases are classified as sporadic (sPD) with no family history, making it challenging to identify specific genes and environmental triggers. PD pathology is characterized by bradykinesia, muscular rigidity, unstable posture and gate, and resting tremor [100]. Additionally, non-motor features including dementia and depression have been described. A plethora of research has demonstrated a link between epigenetic dysregulation and Parkinson’s disease [101]. The initial link between PD and dysregulation of histone acetylation was established from findings that the PD-linked α-syn protein interacts with histones that caused inactivation of HATs that include CBP, p300, and PCAF, triggering enhanced apoptotic cell death in human neuroblastoma cells [102,103]. The A53T mutation in SNCA also significantly decreased acetylation of histone H3 in cultured SH-SY5Y cells and reduced histone H3 acetylation is found in the striatum and substantia nigra of PD mice injected with a-synuclein folded fibrils (PFFs). PFF also reduces CBP/p300-associated factor TADA2A, which is required for acetylation of histone H3/H4 [104]. Additionally, in primary cortex neurons of transgenic mice that express human wild-type α-synuclein, upregulation of α-synuclein is linked with reduced H3K9ac and increased H3K14ac and H3K18ac, where both H3K14 and H3K18 are important histone modifications that are known to facilitate regulation of active transcription [105]. Moreover, brain-specific Tip60 HAT levels are reduced in the *Drosophila* PD model, causing H4K16 and H4K12 acetylation reduction at cognition-linked gene loci with concomitant transcriptional repression [89].

Both HDAC6 and SIRT2 are histone deacetylases (HDACs) that are primarily localized in the cytoplasm and have both been implicated in PD pathogenesis. HDAC6 and SIRT2 both play a role in deacetylation of α -tubulin and additional proteins that mediates microtubule dynamics and trafficking. Multiple findings demonstrate that age-dependent accumulation of SIRT2 in the brain and spinal cord in mice correlates with decreased α-tubulin acetylation in cortical neurons that reduces acetylation of α synuclein, thereby increasing its aggregation and toxicity in neurons [106,107]. Additional research has demonstrated that genetic or pharmacological inhibition of SIRT2 rescued α-synuclein toxicity in dopaminergic neurons. Accordingly, studies showed that reducing SIRT2 activity in human neuroglioma cells inhibits α-synuclein toxicity by promoting enlargement of α-synuclein inclusions [108]. Interestingly, reduced histone acetylation in the PD brain has been shown to be brain-region specific. For example, assessment of histone acetylation in CNS of post-mortem patients with PD and age-matched controls using high resolution studies such as ChIP and RNA-Seq reveals a significant increase in acetylated histone residues in PD patients compared to control individuals. In contrast, the histone acetylation of AcH2A-K3, AcH3-K9, and AcH3-K5 did not increase in the cerebellar cortex in PD patients [109]. Further, studies have identified H3K27-acetylated regions in over 20 genes associated with familial or sporadic forms of PD and are also involved in dysregulating PD associated genes that include SNCA, PARK7, and MAPT [110]. Together, these studies support dysregulation of histone acetylation, alongside inherited genetic mutations and environmental factors, as contributing factors for PD. 

Significant efforts have focused on developing HDAC inhibitors for potential therapeutic use in PD. In both cultured cells and transgenic fly lines, the administration of two different HDAC inhibitors, NaB or SAHA, which showed efficacy for HD, also showed significant protection against α-synuclein-dependent neurotoxicity in PD [102]. For instance, valproic acid (VPA), a class I/IIa HDAC inhibitor, displays neuroprotective effects by increasing acetylation in tyrosine hydroxylase (TH)-positive dopaminergic neurons, while decreasing activated microglia in the striatum, improving motor function and cell survival in PD genetic model mice [111]. NaB was also found to have beneficial effects in brain function and reduced dopaminergic neuron death [112]. However, despite its neuroprotective potential, VPA is associated with various adverse effects, including reduced platelet count or an increased risk of hyperammonemia [113,114]. Notably, it has also caused impairments in patients that mimic PD [115,116]. Patients undergoing long-term VPA therapy reported to have symptoms such as bradykinesia, rigidity, postural instability, and resting tremor [117]. These findings highlight the need for alternative therapeutic strategies, potentially through HAT-targeted therapy to enhance treatment efficacy while minimizing adverse effects.

### 4.3. Amyotrophic Lateral Sclerosis (ALS) 

ALS is defined as a fatal ND disease caused by the deterioration of neurons within the brain cortex, brainstem, and spinal motor neurons resulting in generalized weakness, muscle atrophy and paralysis [118,119]. Prognosis is poor and patients typically succumb to the disorder within three years of diagnosis [118]. ALS can be genetically inherited and familial with autosomal dominant transmission, although 90% of cases are sporadic with no family history. Several genes have been implicated in the pathogenesis of ALS, including superoxide dismutase 1 (*SOD1*), Fused in Sarcoma (*FUS*), chromosome 9 open reading frame 72 (*C9orf72*), and TAR DNA binding protein 43 (*TDP-43*) [120,121]. Histone hypoacetylation has been repeatedly shown to be associated with ALS. For example, the FUS gene that encodes a nuclear Fused in Sarcoma protein, which is associated with motor dysfunction in ALS and has been shown to inhibit CBP/p300 when overexpressed in HeLa cells, causing cell cycle disruption [122,123]. Moreover, due to the conserved cellular pathways between yeast and humans, researchers have found that yeast models overexpressing human FUS exhibit reduction in histone H3 acetylation at lysine residues 14 and 56 (H3K14ac and H3K56ac) [124]. Further, the ELP3 HAT has been linked to ALS-associated motor neurodegeneration in a microsatellite-based genetic association study [125,126]. Using zebrafish and mouse ALS models, other researchers found that expression of human ELP3 was neuroprotective and helped increase longevity and delayed onset of the disease [127]. Additionally, studies using a well-characterized ALS *Drosophila* model reveal brain-specific reduction in Tip60 HAT that resulted in reduced H4K16 and H4K12 acetylation at cognition linked gene loci with concomitant transcriptional repression [89].

Therapeutic restoration of histone acetylation has been explored in ALS by utilizing HDACi [128,129]. ALS has been attributed to gain-of-function mutations in the gene-encoding Cu/Zn superoxide dismutase 1 (*SOD1*) in which *SOD1* point mutation in an ALS mouse model causes motor neuron degeneration [130]. Treatment with HDAC inhibitor, trichostatin A (TSA), reduced motor neuron degeneration, prolonged lifespan, and reduced gliosis in the spinal cord of a SOD1-G93A mouse model [131]. Pharmacological inhibition and/or genetic silencing of HDAC6 rescues axonal transport defects in an ALS-associated mutant FUS mouse model [132]. Perplexingly, gene expression levels for class I, II, and IV HDACs in the ALS brain and spinal cord revealed that mRNA and protein levels of HDAC2 and HDAC11 were activated and repressed, respectively [133]. Accordingly, loss of HDAC function is associated with ALS in some models whereas HDAC inhibition is neuroprotective in others. Thus, future studies are exploring these seemingly opposite acetylation landscapes to elucidate how they contribute to ALS.

### 4.4. Alzheimer’s Disease

Alzheimer’s disease (AD) pathology is hallmarked by a progressive decline in cognition and memory, neuronal cell death, and the accumulation of neurofibrillary tangles and amyloid beta plaques [134]. AD is the leading cause of dementia with 95% of all AD cases arising as late-onset sporadic cases (LOAD), with only 5% being linked to disease-causing genes that include apolipoprotein E (*ApoE4*), presenilin (*PSEN*), and amyloid precursor protein (*APP*) [135]. Sporadic AD is complex, resulting from an interplay of genetics, environmental, and aging conditions. Several common hallmarks have been identified in AD, including the miscleavage of APP into amyloid beta plaques and neurofibrillary tangles (NFTs) in the brain caused by intraneuronal aggregates tau, which is a protein associated with microtubules and neurodegeneration [136]. During early stages of AD progression, neuronal signaling is disrupted and causes mild cognitive impairment (MCI). It is now widely accepted that alterations in histone acetylation with concomitant transcriptional dysregulation are critical contributing factors to MCI-associated early neuronal impairments [122,134,137,138,139]. Evidence supporting an association between histone acetylation and cognitive impairment in AD originally stemmed from observations that histone acetylation was reduced in multiple AD mouse models [78,140]. For example, decreased H4 acetylation but not H3 acetylation was found in a well-characterized tg2576 mouse model for amyloid pathology [141]. Similarly, research using well-characterized *Drosophila* models for AD have shown reduced Tip60 HAT levels in the AD brain with concomitant disruption of histone acetylation marks and cognition-linked gene repression that is recapitulated in the brains of AD patients [47]. Intriguingly, recent studies investigating histone acetylation in specific subregions of the AD brain revealed that histone acetylation dysregulation may differentially affect pathology stages and brain subregions. In support of this concept, researchers researobserved that histone acetylation was differentially altered in AD patients for H3K27ac in post-mortem entorhinal cortex brain region, with most sites depleted in acetylation, while some were enriched [142]. In contrast, researchers found dramatic loss of H4K16ac in the proximity of genes associated with aging, and AD in the lateral temporal lobe of post-mortem AD patients’ brains [143]. It is important to note that H4K16 acetylation has been directly implicated in cognition [47,81]. Accordingly, researchers showed APP-mediated hypoacetylation of H4 at critical learning and memory genes in the prefrontal cortex of mouse brain [144].

A substantial body of evidence shows that inhibition of HDACs can be protective and beneficial in AD. Indeed, APP overexpression in cultured cortical neurons leads to H3 and H4 hypoacetylation, accompanied by a decreased in CBP levels [63]. Additionally, in the hippocampus of APP/PS1 transgenic mice, researchers found a significant reduction in CBP mRNA and protein levels, which resulted in reduced H3 acetylation in hippocampal neurons [145]. Loss of function mutations in genes encoding PS1 and PS2 reduces CBP expression and its target genes such as *c-fos* and *BDNF*, along with increased levels of p25/Cdk5 activator result in negative consequences on spatial and contextual memory and synaptic plasticity [146]. Thus, treatment with the broad HDACi NaB increased H3 and H4 acetylation levels, and caused restoration of learning abilities, including access to long-term memories that was ablated by prior hyperactivation of p25/Cdk5 [74]. Additionally, both general and class I-selective HDAC inhibitors have been demonstrated to ameliorate cognitive defects in an AD mouse model containing a hereditary genetic AD mutation [147,148]. Recent work shows that pan-HDAC inhibitor VPA, as well as WT161, a novel HDAC6 selective inhibitor primarily used in tumor treatment, effectively reduced Aβ deposition in both AD cell and mouse models [134]. These treatments also improved cognitive function in AD mice, further supporting the therapeutic potential of HDACis in AD. Substantial progress has been made in enhancing treatments for NDs, emphasizing the enhancing of selective inhibitor drugs. Among these, HAT-targeted therapies represent a promising avenue, offering potential for more effective clinical translation, and is discussed in the next section.

## 5. HAT Activation as a Promising Therapeutic Strategy for Age-Related Neurodegenerative Disease

It is widely accepted that histone acetylation levels can become inappropriately reduced during neuronal development via reduction in specific HAT activity, with harmful consequences on neuronal activity that culminate in multiple types of neurodegenerative disorders [149] (Table 1). As histone acetylation homeostasis is tilted by inappropriate HAT reduction and HDAC gain, the HAT: HDAC homeostasis ratio tilts in favor of HDACs regarding their availability and enzymatic functionality. While current epigenetic therapies that utilize HDACi treatments to restore histone acetylation have shown promise in reversing cognitive deficits, their therapeutic use is limited due to their non-specific modes of inhibiting multiple HDACs, some of which are critical for cognition. Additionally, HDACi treatments can elicit non-specific chromatin acetylation patterns that might be more harmful than beneficial. For example, superpolyamide hydroxamic acid (SAHA) and Nicotinamide (vitamin B3) have been advanced to Phase I and Phase II clinical trials as representative HDAC inhibitors for AD treatment [150,151]. Despite the initial promise, their poor blood–brain barrier (BBB) permeability, lack of specificity by targeting several HDACs, and low HDAC inhibitory potency, make these pan-HDAC inhibitors face limited and negative outcomes because they produce global hyperacetylation and widespread transcriptional dysregulation in clinical trials [152,153,154]. Similarly, SAHA is a pan-inhibitor that targets class I and class II HDACs (HDACs 1-10), and it has been shown to have positive results in AD animal models. However, despite its demonstrated efficacy in animal models, determining an optimal dose–response curve in clinical trials remains a significant challenge, thereby limiting its therapeutic potential in humans [152]. Similarly, Nicotinamide, another pan inhibitor of all class III sirtuin HDACs (SIRT1-7), failed to demonstrate significant cognitive outcomes in Phase II clinical trials involving patients with mild to moderate AD [155]. Additional evidence suggests that Nicotinamide may exacerbate several neuropathological features in mouse models of PD, which further raises concerns about the use of this inhibitor’s safety profile in other neurodegenerative contexts [156].

In contrast, targeting HATs over HDACs presents benefits since HATs, unlike HDACs, play non-redundant critical roles in “re-writing” the specific chromatin histone acetylation patterns required for cognition gene expression and higher-order brain function. For example, reduced Tip60 protein levels are observed in the brains of human post-mortem AD hippocampus and in a well-characterized AD *Drosophila* model, causing decreased Tip60-mediated site-specific cognition-associated histone acetylation marks with concomitant repression of critical neuroplasticity genes [47]. Importantly, genetically increasing Tip60 in the *Drosophila* AD brain prevents site-specific histone hypoacetylation and restores activation of repressed neuroplasticity genes [47,157]. Increasing Tip60 levels also restore multiple neural processes impaired in early stages of AD, including axonal outgrowth/transport, learning/memory, circadian rhythm, synaptic plasticity, locomotor function, apoptotic driven neurodegeneration, and neuroplasticity gene control [16,22,43,46,47,75,89,158,159]. Together, these studies support an overall neuroprotective role for Tip60 in multiple neural cognitive circuits impaired in AD. Thus, activation of specific HAT using therapeutic modulators can potentially have more direct and causative positive effects for treatment of NDs.

Although HDACi treatment induces general hyperacetylation in multiple NDs, there are additional limitations to this treatment in that it is often not sufficient to produce absolute beneficial effects alone and requires restoration of the specific HATs that “write” the site-specific acetylation marks. For instance, increased p300, but not HDAC inhibition, promotes axonal regeneration in mature retinal ganglion cells after optic nerve injury. These beneficial effects are mediated by p300-induced hyperacetylation of histone H3 and p53, causing increased expression of selected pro-axonal outgrowth genes [160]. Further, additional research demonstrated that general induction of histone acetylation by HDACi treatment alone failed to rescue synaptic plasticity and memory loss in the hippocampus of a mouse model lacking the learning and memory-linked HAT CBP, and that a wild-type CBP allele in conjunction with HDACi treatment was required [161]. Such incomplete rescue by HDACi treatment alone is likely due to the necessity of CBP for site-specific histone acetylation and the recruitment of basal transcriptional machinery required for the activation of key synaptic plasticity target genes. Consistent with these findings, other researchers showed that enhancing Histone H4 acetylation by using the HDAC inhibitor, MS-275, in a *Drosophila* AD-associated APP model was insufficient to fully restore the expression of certain Tip60 target genes [16]. This is likely since Tip60 is still required for site-specific acetylation and interactions with other transcription factors. In summary, these support the promising concept that enhancing the activity of specific HATs like CBP or Tip60 might serve as alternative therapeutic strategy for alleviating cognitive impairments that characterize multiple neurodegenerative disorders.

Importantly, certain HATs also play critical roles in non-chromatin-mediated cellular processes critical for cognitive function that can also be exploited in ND therapeutics by specific HAT activation (Table 2). For example, Elp3 HAT acetylates microtubules critical for synaptic bouton increase during neurogenesis [125]. More recent evidence demonstrates that disruptions in ELP3-mediated microtubule acetylation are a common feature in multiple neurological disorders, presenting ELP3 action as a potential target for acetylation-based therapeutic interventions [162]. Similarly, *Drosophila* Tip60 functions in regulating synaptic plasticity in the neuromuscular junction, in part via microtubule acetylation [158]. More recently, it was reported that Tip60 carries out a novel RNA-binding function in addition to its HAT function that modulates alternative splicing of pre-mRNA targets implicated in AD [163]. Indeed, defects in alternative RNA splicing have emerged as a widespread mechanism in multiple neurodegenerative diseases including AD, bringing another layer of complexity to solving the neurodegenerative disease puzzle. Alterations in RNA splicing culminate in aberrant protein isoform variations that interfere with neuronal cellular processes. Intriguingly, Tip60 HAT action and RNA splicing function may be co-dependent such that modulating one function may also have an impact on the other function. Although the precise mechanism by which Tip60 targets RNA is yet to be elucidated, Tip60 RNA immunoprecipitation (RIP) and chromatin immunoprecipitation (ChIP) studies reveal an extensive overlap between Tip60’s gene targets at the chromatin and RNA level. These results support a model by which Tip60-RNA targeting occurs very close to its target chromatin loci [163]. Tip60’s specificity for pre-mRNA may arise from its proximity to the new transcripts. After Tip60 is recruited it to specific chromatin loci for HAT mediated gene activation, it can then switch to the newly synthesized pre-mRNA strand to control its splicing. Thus, increasing Tip60’s HAT action in NDs such as AD using Tip60-specific HAT activators might also restore the Tip60 RNA splicing function due to enhancement of transcriptional activation that might also increase Tip60’s RNA-targeting and splicing function [163].

Importantly, activation of specific HATs using therapeutic modulators has been shown to have direct and causative positive effects for treatment of NDs. For example, CBP/P300 activators CSP-TTK21 and CTPB, and recently discovered Tip60 HAT activators can cross the blood–brain barrier (BBB) to promote neuroprotection in multiple ND animal models [12,13,164]. The HAT activator CSP-TTK21 was demonstrated to promote adult neurogenesis and spatial memory formation with significant increases in histone acetylation within the hippocampus and frontal cortex in adult mice [12]. Additionally, CTPB was shown to have neurotrophic effects when administered to SH-SY5Y cells to study PD, enhancing cell survival and growth [13]. Notably, a recent study by Bhatnagar et al. (2025) [164] identified novel small molecule Tip60 HAT activator compounds that exhibited efficacy in rescuing functional neuronal deficits in vivo in both Tip60 knockdown *Drosophila* and well-characterized AD-associated APP *Drosophila* models. These compounds also restored repressed plasticity gene expression and significantly extended lifespan in AD flies. These exciting findings represent a significant advance over previously explored pan-HDACi treatments and highlight the potential for the use of HAT activators as a therapeutic strategy for multiple NDs. 

## 6. Conclusions

In conclusion, histone acetylation is currently highlighted as a critical mechanism that controls gene regulatory programs required for cognition. Although tightly regulated histone acetylation and deacetylation is critical for maintaining proper neuronal function, age-related dysregulation leads to alterations in the neuroepigenetic environment that impairs cognitive abilities. Dysregulation of histone acetylation chromatin writers can lead to cognitive deficits, neuronal apoptosis, and neurodegeneration, all hallmarks of ND pathology. Future research will be critical in unraveling the precise mechanisms underlying how specific HATs positively impact cognition and slow or prevent neurodegeneration in different age-associated NDs. Indeed, the differences in mechanisms underlying HAT/HDAC contribution to AD, PD, HD, and ALS, along with the varying types of histone acetylation alterations in different subregions of the brain and at various stages of disease progression observed in the multiple types of NDs, highlight the inadequacy of a one-size-fits-all therapeutic strategy. Thus, further investigation into disease stage and brain region-specific acetylation landscapes between multiple NDs is necessary for advancing the development of HAT-specific therapeutic strategies with improved precision and efficacy. This knowledge will be informative for developing specific HAT activation-targeted therapeutic strategies in the early intervention of NDs. Furthermore, there are limitations that must be noted regarding any drug discovery, since the development of compounds routinely involves multiple rounds of testing and optimization before finding the most clinically effective compounds. Throughout the drug discovery process, it is also favorable to select for optimal pharmacokinetic properties, when feasible, to design drugs that display favorable absorption, distribution, metabolism, and excretion (ADME) characteristics, together with preferential blood–brain barrier (BBB) transport. Thus, future directions for HAT activator drug discovery involve lead compound optimization that would then necessitate testing for in vivo pharmacokinetics and efficacy in mammalian models of ND. Additionally, since NDs are complex, HAT activators may be more suitable as a combinatorial therapy in conjunction with genetic and/or other neuroepigenetic modulator therapeutic strategies. In summary, exploration into the generation of small molecular compounds that selectively target and enhance the activity of neuroprotective HATs like CBP, p300, and Tip60 should facilitate the development of HAT-based therapeutics with reduced adverse effects for patients afflicted with these debilitating neurodegenerative disorders.

## Figures and Tables

**Figure 1 cells-14-00873-f001:**
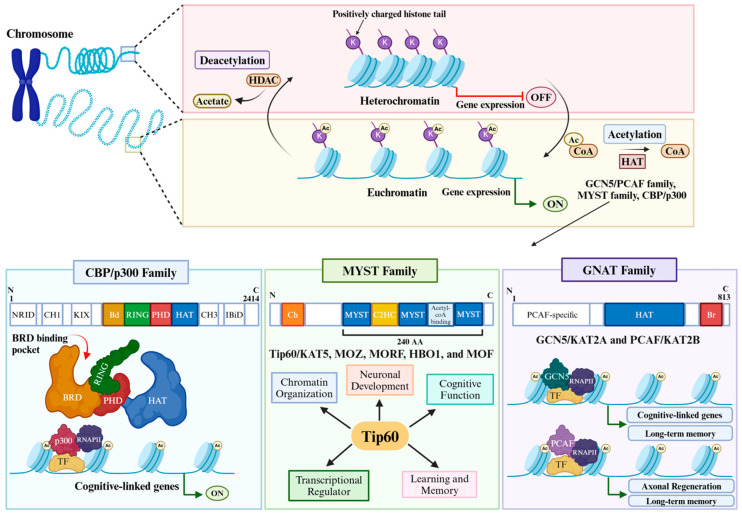
HAT families in brain function and cognition. The three most notable HAT families that are required for brain function and cognition are the CBP/p300, MYST, and GNAT. Their functional roles in the specific cognition-linked neuronal cellular processes they control are illustrated.

**Figure 2 cells-14-00873-f002:**
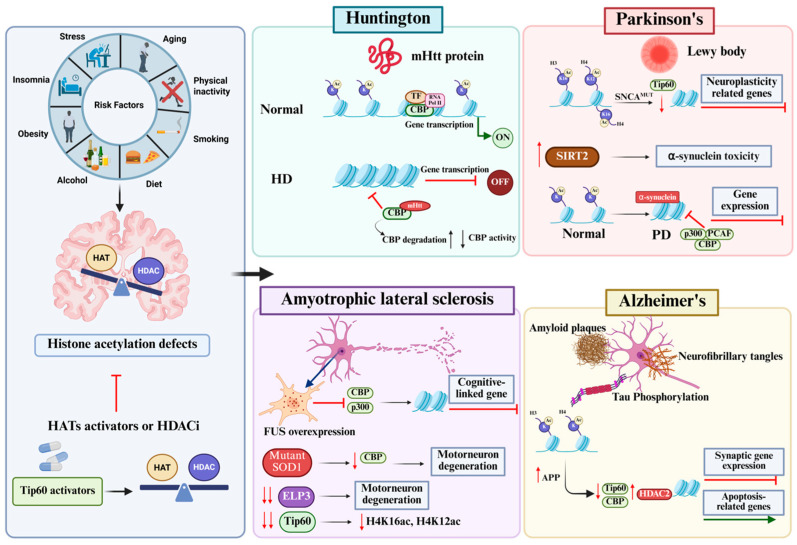
HAT-mediated neuroepigenetic alterations drive multiple age-related neurodegenerative diseases. Reduced HAT and histone acetylation levels are found in the brains of animal models for multiple types of neurodegenerative diseases. Pharmacological treatments aimed at increasing histone acetylation by inhibiting HDAC action or activating specific HATs in these models have shown reversal of cognitive deficits and are thus a topic of intense research. This schematic provides an overview of data linking impaired control of specific HATs and HDACs to alterations in the neuroepigenome that contribute to multiple neurodegenerative diseases.

**Table 1 cells-14-00873-t001:** HAT/HDAC modulation and dysregulation within neurodegenerative diseases.

HAT/HDAC	ND	Modulation	Reference
CBP (HAT)	Huntington’s	Redistributed in cytoplasm vs nuclear	[86]
		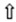 ubiquitylation and degradation	[87,91,92]
		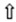 Inhibition caused by inappropriate binding of first exon of HTT	[88]
	Parkinson’s	Inactivation caused by alpha-syn protein	[102,103]
		Unaltered levels	
	Amyotrophic Lateral Sclerosis	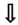 Inhibition caused by overexpression of FUS gene	[63,123,130]
	Alzheimer’s	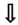 levels by loss of function mutations in PS1 and PS2	[146]
PCAF (HAT)	Huntington’s disease	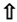 Inhibition caused by inappropriate binding of first exon of HTT	[87,88]
	Parkinson’s	Inactivation caused by alpha-syn protein	[102,103]
P300 (HAT)	Parkinson’s	Inactivation caused by alpha-syn protein	[102,103]
	Amyotrophic Lateral Sclerosis	Inhibition caused by overexpression of FUS gene	[122,123]
SIRT2 (HDAC)	Parkinson’s	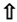 levels and accumulation	[108]
	Huntington’s	Suppression leads to neuroprotection	[93]
ELP3 (HAT)	Amyotrophic Lateral Sclerosis	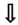 levels	[127]
Tip60 (HAT)	Huntington’s	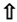 levels	[157]
	Parkingson’s	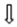 levels	[157]
	Amyotrophic Lateral Sclerosis	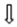 levels	[157]
	Alzheimer’s	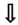 protein levels	[47]
RPD3 (HDAC)	Huntington’s	Suppression leads to neuroprotection	[93]

**Table 2 cells-14-00873-t002:** Therapeutic Treatments: HDAC inhibitors and HAT Activators.

Therapeutic	Modulation	ND Applicable	Clinical Application	Reference
TSA	Small molecule HDAC Inhibitor	Huntington	Pre-clinical/experimental	[90]
		ALS	Pre-clinical/ experimental	[131]
Vorinostat (SAHA)	Small molecule pan-HDAC inhibitor	Huntington	Pre-clinical/experimental	[90]
		Parkinson’s	Pre-clinical/ experimental	[102]
		Alzheimer’s	Pre-clinical/experimental	[147]
Sodium Butyrate	Short fatty acid HDAC inhibitor	Alzheimer’s	Pre-clinical/experimental	[147]
		Parkinson’s	Pre-clinical/experimental	[112]
		Huntington	Pre-clinical/experimental	[102]
Sodium Valproate (VPA)	Small molecule HDAC inhibitor Negatively impacts Acetyl-CoA action	Parkinson’s	Pre-clinical/experimental	[111]
		ALS	Pre-clinical/experimental	[63]
		Alzheimer’s	Pre-clinical/experimental	[134,147]
4B	Pimelic diphenylamide HDAC inhibitor	Huntington’s	Pre-clinical/experimental	[95]
RGFP966	Selective HDAC 3 inhibitor	Huntington’s	Pre-clinical/ experimental	[96]
CSP-TTK21	P300/CBP Hat activator	Alzheimer’s	Pre-clinical/experimental	[12,13]
CTPB	HAT Activator	Alzheimer’s	Pre-clinical/experimental	[13]
P-compounds (P10, P13)	Specific Tip60 HAT Activator	Alzheimer’s	Pre-clinical/experimental	[163]
WT161	Selective HDAC6 inhibitor	Alzheimer’s	Preclinical/ experimental	[134]

## Data Availability

Not applicable.

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
