# Peer review of "Modulating Cognition-Linked Histone Acetyltransferases (HATs) as a Therapeutic Strategy for Neurodegenerative Diseases: Recent Advances and Future Trends"

_cells, 2025, doi:10.3390/cells14120873_

Round 1
Reviewer 1 Report
Comments and Suggestions for Authors
General Assessment:
The manuscript titled "Modulating cognition-linked HATs as a therapeutic strategy for neurodegenerative disease: recent advances and future trends" is an insightful and well-structured review that significantly contributes to the field of neuroepigenetics. It effectively synthesizes recent findings on the role of histone acetyltransferases (HATs) in cognition and their therapeutic potential for neurodegenerative diseases across various model systems. However, several areas require substantial revision to enhance clarity, readability, and overall impact. These include addressing typographical errors, improving flow and readability, balancing the depth of sections, and strengthening the conclusion with specific future directions and limitations.
Specific Comments by Section:
Title: While the title is concise and focused, spelling out "histone acetyltransferases" alongside "HATs" would improve accessibility for readers unfamiliar with the acronym. For example: "Modulating Cognition-Linked Histone Acetyltransferases (HATs) as a Therapeutic Strategy for Neurodegenerative Diseases."
Abstract: Simplifying terms and improving transitions between pathology and therapeutic applications would enhance flow and broaden readership.
Keywords: The keywords should not use abbreviations and must align with MeSH terms. Replace "HATs" with "Histone Acetyltransferases" and "NDs" with "Neurodegenerative Diseases." Ensure all keywords are MeSH-compliant to improve searchability and consistency.
Introduction: The introduction establishes a strong foundation but could benefit from a concrete example, such as "reversible acetylation altering memory gene expression," to make it more engaging. Additionally, simplifying complex sentences and breaking them into shorter, digestible parts will enhance readability and accessibility for a broader audience.
HAT Families in Brain Function: This section is comprehensive and well-supported with evidence from diverse models. However, a brief introductory summary would provide helpful context and prepare readers for the technical depth of the discussion.
HAT:HDAC Interplay: The discussion of gene regulation mechanisms is insightful but could be made more accessible to less specialized readers. Clarifying how neural cues drive acetylation patterns would aid understanding and ensure broader appeal.
Neuroepigenetic Alterations : This section is rich in content but uneven in its coverage. Huntington’s and Alzheimer’s diseases are discussed comprehensively, while Parkinson’s and ALS need expansion. For example, additional examples or unresolved questions about ALS acetylation discrepancies would add depth and balance to the discussion.
Therapeutic Modulation: The innovative emphasis on HAT specificity over HDAC inhibitors is a highlight of the review. However, briefly noting practical challenges, such as drug delivery and blood-brain barrier penetration, would balance optimism with realism and provide a more nuanced perspective.
Conclusion: The conclusion effectively summarizes the key themes of the review but would benefit from greater specificity and a stronger forward-looking perspective. To enhance its impact, the authors should incorporate explicit research gaps, such as the need for selective HAT activators (e.g., Tip60-targeting small molecules) and a deeper exploration of disease-specific acetylation inconsistencies. Limitations, including challenges like drug specificity and blood-brain barrier penetration, should be acknowledged to provide a balanced view. Additionally, a clear call to action, such as encouraging future studies to address unresolved questions in ALS or Parkinson’s disease, would strengthen the conclusion’s clinical relevance and forward momentum. This revision would ensure the conclusion not only reflects the review's contributions but also provides actionable insights for advancing the field.
General Recommendations:
-It is suggested that a table summarizing the effects of HAT modulation for each disease (Huntington’s, Parkinson’s, ALS, Alzheimer’s) be created to consolidate findings and enhance readability. This would provide readers with a clear overview of the therapeutic impacts discussed across different neurodegenerative conditions.
-To improve transparency and credibility, it is recommended that the authors include a brief note on the literature selection process, even though reviews typically lack a formal "Methods" section. For instance, a statement such as the following could be added: "This review synthesizes peer-reviewed studies from 2015–2025, sourced via PubMed, Scopus, and Google Scholar using keywords like ‘HATs,’ ‘histone acetylation,’ ‘neurodegenerative disease,’ and ‘cognition.’ Priority was given to mechanistic and therapeutic insights, while excluding non-primary or unrelated works." Specifying the exact keywords and search engines used would clarify the approach and allow others to replicate the process.
-Additionally, it is advised that each disease be briefly defined using authoritative sources to ground the discussion. These definitions could be integrated throughout Section 4, and references should be updated accordingly. Suggested papers for this purpose include:
- Unveiling the role of histone deacetylases in neurological diseases: focus on epilepsy. Biomarker Research, 12(1), 142. doi: 10.1186/s40364-024-00687-6
- Multi Targeted Therapy for Alzheimer's Disease by Guanidinium-Modified Calixarene and Cyclodextrin Co-Assembly Loaded with Insulin. ACS Nano, 18(48), 33032-33041. doi: https://doi.org/10.1021/acsnano.4c05693
- Quercetin-functionalized nanomaterials: Innovative therapeutic avenues for Alzheimer's disease management. Ageing Research Reviews, 104, 102665. doi: https://doi.org/10.1016/j.arr.2025.102665
- TGF-β1 mediates hypoxia-preconditioned olfactory mucosa mesenchymal stem cells improved neural functional recovery in Parkinson’s disease models and patients. Military Medical Research, 11(1), 48. doi: 10.1186/s40779-024-00550-7
- Apelin-13 suppresses neuroinflammation against cognitive deficit in a streptozotocin-induced rat model of Alzheimer’s disease through activation of BDNF-TrkB signaling pathway. Frontiers in pharmacology, 10, 395. doi: https://doi.org/10.3389/fphar.2019.00395
- Untargeted metabolomics analysis of the hippocampus and cerebral cortex identified the neuroprotective mechanisms of Bushen Tiansui formula in an aβ25-35-induced rat model of Alzheimer's disease. Frontiers in Pharmacology, 13, 990307. doi: 10.3389/fphar.2022.990307
-References: The reference list is strong but requires minor cleanup, such as addressing truncated entries and ensuring consistent DOI formatting. Additionally, the authors can enhance the manuscript by incorporating recently published papers to reflect the latest advancements in the field.
-Proofreading and Clarity: It is recommended that typos be addressed and dense or awkward phrasing refined to improve readability. Breaking complex sentences into shorter, digestible parts could enhance accessibility for a broader audience.
-Text Match (59%): If this percentage indicates literature overlap, it is typical for a review article. However, it is suggested that unique phrasing, particularly in therapeutic sections, be emphasized to enhance originality. Clarification of its significance, such as in the context of a plagiarism check, could be beneficial if relevant.
-List of Abbreviations: It is advised that a list of abbreviations be provided after the conclusion to ensure clarity and ease of reference for readers.
Reviewer 2 Report
Comments and Suggestions for Authors
Mai and co-workers summarized the recent advances concerning the modulation of HATs as possible therapies in neurodegenerative disorders. With the aging of societies, neurodegenerative diseases become more and more widespread, and usually only symptomatic treatments are available, but disease-modifying therapies are missing. Therefore, the review is timely and of great interest. It gives mostly a detailed insight into the topic, however, there are some questions/concerns.
Major question/concerns:
The figures are really informative, but the resolution of the figures may be improved for better legibility.
HDAC6 and SIRT2 are often mentioned in the text related to their role in various neurodegenerative disorders. Nevertheless, it is mentioned very briefly that these enzymes have multiple substrates. I would suggest a little more details about their role. Especially, that these two HDACs are mainly localized in the cytosol and can be found in the nucleus in much less extent. Moreover, these two enzymes have several substrates, for example tubulin and alpha-synuclein. Their role in neurodegeneration was found to be important, but usually by their functions affecting the microtubule network (HDAC6 and SIRT2) and even directly deacetylate SYN (SIRT2) in Parkinson’s disease (Cappelletti et al. Neurosci Lett. 2021 Jun 11;755:135900. doi: 10.1016/j.neulet.2021.135900. Epub 2021 Apr 18. PMID: 33878428; de Oliveira et al. PLoS Biol. 2017 Mar 3;15(3):e2000374. doi: 10.1371/journal.pbio.2000374. eCollection 2017 Mar. PMID: 28257421; Esteves et al. Biochim Biophys Acta Mol Basis Dis. 2019 Aug 1;1865(8):2008-2023. doi: 10.1016/j.bbadis.2018.11.014. Epub 2018 Dec 17. PMID: 30572013). It is difficult to judge the extent of their activity on histones and the other substrates in cytoplasm.
In the cases of most of the examples, TSA or SAHA was used as inhibitors. These inhibitors have broad specificity, they usually inhibit not a given HDAC, but most of them (class I/II).
line 448-449: I agree that dysregulation of histone acetylation contribute to the pathomechanism of Pd, but I am not sure that it is the “molecular cause” of the disease.
Section 5 was interesting, but I sorely miss a table containing the known HAT modulators. Are there small molecule activators? What other methods are available to modulate HAT activities? Have any clinical trials been conducted with such methods?
Minor remarks:
There is some misspelling or not appropriate formatting:
See Fig1. legend: in rain function instead of in brain function.
lines 329, 579-581, 598-605.
Round 2
Reviewer 1 Report
Comments and Suggestions for Authors
The authors have addressed the concerns raised, and the manuscript is now suitable for publication, but a final plagiarism check is required due to the initial high duplication rate.
Reviewer 2 Report
Comments and Suggestions for Authors
The authors have answered my questions and comments.